# Video Language Models are Human-Aligned Evaluators for Text to Motion Generation

## Abstract

Recently, text-to-motion (T2M) has become a basic setting for human motion generation. This work studies the evaluation of alignment between text and generated motion, to credit the reliable use of T2M models. We consider solving the T2M evaluation task by making use of a video language model (VLM). Our basic idea is: render the generated human motion into a skinned video, and then use a VLM for evaluation. To address information loss problem when 3D motion is rendered into 2D video, we develop a method, which ensures reliable evaluation score by analyzing VLM entropy. Our evaluation method, named VeMo, frees T2M evaluation from reliance on motion data while seamlessly leveraging the semantic understanding and reasoning capabilities of advanced VLMs trained on Internet-scale data. To systematically compare the empirical usefulness of different evaluation methods, we manually annotate a meta-evaluation benchmark that includes coarse-grained alignment labels and fine-grained judgmental reasons. Extensive experiments and case studies demonstrate the effectiveness of the proposed VeMo.

## 1 Introduction

Text-to-motion (T2M) (Obludzyner et al., 2024) has emerged as a foundational setting for human motion generation, where the objective is to produce continuous human motion sequences from free-form natural language descriptions. This task underpins diverse practical applications, such as humanoid robot control, video game character animation, and virtual reality interactions, driving increasing research attention in T2M model development in recent years (Sahili et al., 2025).

Evaluating generated motions is essential for advancing T2M research, as reliable assessment supports model improvement across all generative tasks. Traditional metrics (e.g., FID, L1 distance) focus on comparing generated motions with reference motions. However, one text prompt can map to multiple valid motion sequences, reducing the effectiveness of these metrics. To address this challenge, recent work (Tevet et al., 2022; Voas et al., 2023; Wang et al., 2024) has pretrained evaluation models on text-motion pairs to assess the generated motions in a reference-free manner. However, the high cost of acquiring high-quality data hinders the evaluators' ability.

Compared with motion data, vision data is much easier to acquire (Radford et al., 2021; Xu et al., 2021). Pretrained vision-language models have been used for evaluating the generated images and videos, and exhibit high correlation with human judgments (Tu et al., 2024; Liu & Zhang, 2025). We consider solving the T2M evaluation task by making use of a video language model (VLM). Our basic idea is: render the generated human motion into a skinned video, and then use a VLM for evaluation. To address the problem of information loss caused by issues such as severe human body self-occlusion when 3D motion is rendered into 2D video, we develop a method, which ensures reliable evaluation score by analyzing the entropy of the VLM. Our evaluation method, named VeMo, frees T2M evaluation from reliance on motion data while seamlessly leveraging the semantic understanding and reasoning capabilities of advanced VLMs trained on Internet-scale data.

To systematically compare the empirical usefulness of different evaluation methods, we established the first test-only benchmark. We manually annotate 1101 diverse prompts from the HumanML3D (Guo et al., 2022) test set, using two widely adopted T2M models (MDM (Tevet et al., 2022), MotionGPT (Jiang et al., 2023)) to generate motions, which yields 2202 text-motion pairs for annotation. This benchmark includes: 1) coarse-grained alignment labels (denoting overall text-motion match) and 2) fine-grained judgment labels (e.g., Faithfulness and Naturalness). We also design

Figure 1: Examplified descriptions (left), paired motion frames (middle), evaluation scores (right). On the right side, human-aligned evaluation scores are marked in green, otherwise in red.

a pipeline (e.g., regenerate controversial motions) to ensure full consistency in oracle annotations. The Inter-Annotator Agreement (Krippendorff's Alpha) between oracle annotations and untrained users' annotations exceeds 0.67 (average by label types), demonstrating high data quality.

We compared VeMo with classic reference-based (Tevet et al., 2022) and recent reference-free evaluation methods (Voas et al., 2023; Wang et al., 2024). On our benchmark, VeMo shows the best correlation with human judgments. This confirms that VeMo can provide reliable text-motion alignment assessment without using motion data and human labels, thus addressing existing limitations. To get a qualitative sense, see Fig. 1. We summarize our contributions as follows:

- We study the use of VLMs to evaluate alignment between text and generated motion, enabling internet-scale data to benefit T2M evaluation without the need for T2M data.

- We present a meta-evaluation benchmark to assess prior metrics and our strategy (i.e., video-language models as evaluators), while also incorporating user study.

- We show that VeMo outperforms existing metrics in evaluating T2M alignment.

## 2    RELATED WORK

**Text-to-motion generation** (T2M) aims to to create human motion sequences from free-form natural language descriptions. Recent advances in T2M have centered around two model families: One is discrete-token (VQ-VAE + autoregressive/LLM-based) methods. (Zhang et al., 2023a; Lou et al., 2023; Jiang et al., 2023; Guo et al., 2024; Chen et al., 2024; Li et al., 2025) These methods discretize motion into tokens with a VQ-VAE (Van Den Oord et al., 2017), then generate token sequences through a transformer (Vaswani et al., 2017) or language model (Brown et al., 2020) conditioned on text. Finally, the generated motion token sequence is decoded back into continuous motion using the VQ-VAE decoder. Another research line focuses on continuous latent-space diffusion. (Chen et al., 2023; Shafir et al., 2023; Zhang et al., 2023b; Tevet et al., 2022; 2024; Uchida et al., 2025) These models bypass quantization by learning diffusion dynamics directly in a continuous latent space. Emerging methods such as MoMADiff (Zhang et al., 2025) and LEAD (Andreou et al., 2025) combine discrete and continuous strategies for finer control. The significant strides of T2M models suggest that the evaluation of generated motion is a timely consideration.

**Text-to-motion evaluation.** Traditional metrics rely on reference motions. FID calculates scores for each model by comparing distributions of generated and reference motions, rather than for each text-motion pair, which is excluded from the main baselines and analyzed in Appendices (Table 7). L1 distance measures the distance between each pair of generated and reference motions. Yet, one prompt can map to multiple valid motions, reducing the effectiveness of these metrics. To get rid of the reference, Tevet et al. (2022; 2024); Han et al. (2025) measure the **Multimodal Distance** between text and motion embeddings. **MoBERT** Voas et al. (2023) trains an evaluation model using fine-grained text-motion labels. *However, there are no coarse-grained alignment labels.* **Motion-Critic** (Wang et al., 2024) integrates human perception on generated motions to train an evaluation model. However, MotionCritic mainly studies the quality of the generated motion independent of text. Overall, these methods' generalization is constrained by limited text-motion data. VeMo enables internet-scale text-vision data to benefit T2M evaluation without the need for motion data.

Table 1: Comparison of related work in terms of critic model and human annotated dataset. T denotes the text modality and M denotes the motion modality. "-" means no such resources.

|  | Evaluation Model | | | Generated M w/ Human Label | | |
| --- | --- | --- | --- | --- | --- | --- |
|  | Input | M-format | Trained on M | Input | Test only | Label granularity |
| Multimodal Distance | T,M | Fixed | ✓ | - | - | - |
| MoBERT | T,M | Fixed | ✓ | T,M | ✗ | Fine |
| MotionCritic | M | Fixed | ✓ | M | ✗ | Coarse |
| VeMo (Ours) | T,M | Any | ✗ | T,M | ✓ | Fine → Coarse |

**Meta-evaluation** benchmarks, dedicated resources for comparing empirical usefulness of different evaluation methods, have become foundational in mature generative tasks such as text-to-text/image (Tu et al., 2024; Stufflebeam, 2011; Son et al., 2024). In the field of T2M, while prior works (Voas et al., 2023; Wang et al., 2024) have introduced human-labeled datasets of generated motions, these datasets were used to train their respective evaluation models, not designed to validate the evaluative generalizability of different evaluation methods. This lays the research gap that our benchmark specifically addresses: We compare the empirical usefulness, i.e., generalizability of different evaluation methods by making human labels unseen to them, that is, not providing trainset. Table 1 features the most related evaluation models and human-labeled datasets of generated motions.

## 3 DATASET FOR META EVALUATION

We first collect text and generated motions from existing resources (Sec. 3.1); Then we design a pipeline to collect human annotations (Sec. 3.2). The overall pipeline is depicted in Figure 2(a).

### 3.1 COLLECT TEXTUAL DESCRIPTIONS AND GENERATED MOTIONS

**Data source.** HumanML3D (Guo et al., 2022) is a recent dataset, textually re-annotating motion capture from the AMASS (Mahmood et al., 2019) and HumanAct12 (Guo et al., 2020) collections. It contains 14,616 motions annotated by 44,970 textual descriptions, split in train, val, test sets. The train and val splits of HumanML3D are widely adopted to train T2M models. We take the descriptions from the HumanML3D's test set as prompts to generate and evaluate motions. To ensure the diversity and representativeness of the prompts, we used an advanced Sentence Transformer to remove duplicates from the prompts through hierarchical clustering, resulting in approximately 1.5k prompts. Details of the deduplication can be found in the **Appendices**. Finally, we customized more conditions to further filter the remaining 1.5k prompts. The removal conditions include spelling errors in the action descriptors in the prompts, or prompts describing dexterous hand movements, gaze, and other actions that do not belong to the HumanML3D joints. In the end, 1101 texts remained.

**Motion generation.** A trained T2M model will take textual motion annotations as input and output motion sequence $M = (m_t)_{t=1}^N$ of human poses represented by joint rotations or positions $m_t \in \mathcal{R}^{J \times D}$. $J$ is the number of joints and $D$ is the dimension of the joint representation. Specifically, we employ a diffusion-based MDM (Tevet et al., 2022) and an autoregressive MotionGPT (Jiang et al., 2023) to generate motion data from 1101 selected prompts for subsequent meta-evaluation. Because the codebases of these two models are widely adopted as the foundation for other methods Tevet et al. (2024); Han et al. (2025), and both models are trained on the HumanML3D's trainset and support the animation of body actions for the 22-joint SMPL human model. Finally, we obtain generated motions from MDM and MotionGPT, and there are a total of 2202 text-motion pairs.

**Objective.** After obtaining each pair of text $T$ and generated motion $M$, an evaluation system $\phi$ needs to take $T$ and $M$ as inputs and convert them into a scalar $\phi(T, M)$, which reflects the degree of alignment between $T$ and $M$. The ideal $\phi(T, M)$ is expected to correlate with human annotation.

### 3.2 VISUALIZE MOTION DATA FOR HUMAN ANNOTATION

To annotate a generated motion sequence $M$ and the text used for generation, we first use Blender (Community, 2018) software to convert the generated motion $M$ into skinned human model video $V$. We optimized the rendering environment and camera movement to ensure that the human model's

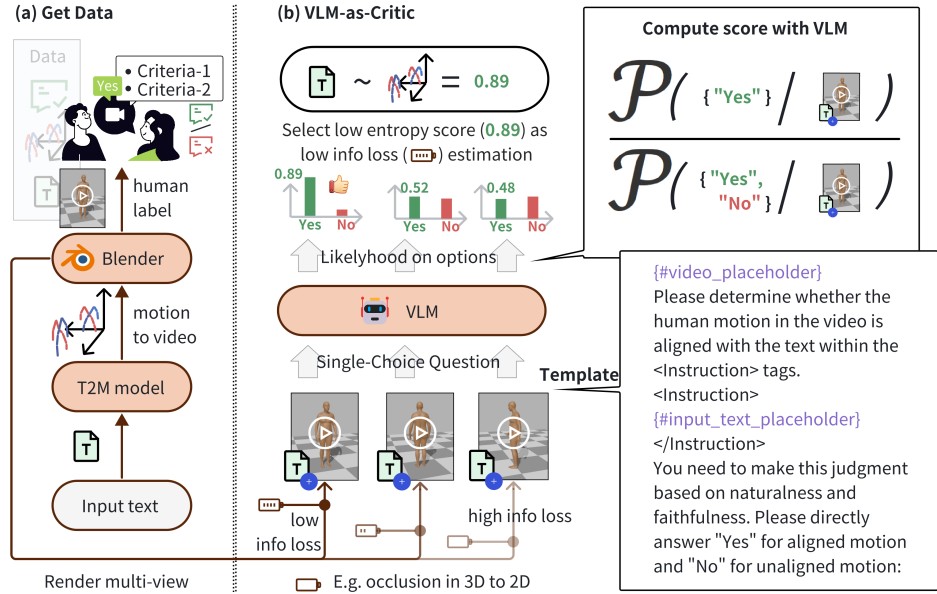

Figure 2: (a) Steps for collecting benchmark data. (b) Framework of our automatic metric: render one motion into videos from multiple views and use video-language model as evaluator.

Table 2: Class weight on two data splits and inter-annotator agreement for different labels.

|  | Alignment | Faithfulness | Naturalness |
|---|---|---|---|
| Positive ratio (MDM) | 613/1101 | 621/1101 | 1069/1101 |
| Positive ratio (MotionGPT) | 506/1101 | 528/1101 | 1046/1101 |

entire body is fully visible in the video, with clear movements, and that the model itself occupies more than 1/10 of the frame, more details can be found in **Appendices**.

We also implement a strict process to ensure annotation quality. Specifically, we designed our human annotation collection to use single-choice questions, as selection is generally easier and more reliable than direct rating (Kendall, 1948; Wang et al., 2024). Given a text and a rendered video $V$ as materials, we instructed postgraduate students as oracle annotators to select one option per fine-grained criterion, which defines the reason why a text-motion pair is mismatched.

- **Text-Motion Faithfulness**: Does the human in the video execute the action depicted in the text completely, accurately, and in the correct order? Select from "Yes" or "No".
- **Motion Naturalness**: Is the human motion in the video natural, without joint distortion or strange movements that go beyond the text description? Select from "Yes" or "No".

Specifically, oracle annotators need to recheck cases with inconsistent annotations, unify the annotation results through discussion, or regenerate motions for annotation — until consistent oracle annotations are obtained for all data. The dataset's statistical information is shown in Table 2. We also conducted user study in the experimental section, where we invited participants with at least a bachelor's degree to independently perform annotations based on the above criteria, and we reported the Inter-Annotator Agreement (IAA) with the oracle annotations in Section 5.4.

The amount of unnatural data is rare. We aggregate the fine-grained judgmental reasons into coarse-grained Alignment labels: A text-motion pair is considered "aligned" only when both its Faithfulness and Naturalness labels are marked as "Yes"; otherwise, it is deemed "unaligned". Finally, we obtained more balanced Alignment labels, which indicate whether the generated motions faithfully and naturally match the prompt. Note that prior evaluation model (Voas et al., 2023) is developed on human labels for Faithfulness and Naturalness. Our datasets can serve as a test set to study its empirical usefulness, i.e., generalizability. Although our annotation pipeline supports more criteria, in this work, we only consider the above, which we believe is the most important.

## 4 VIDEO LANGUAGE MODEL AS EVALUATION MODEL

We first formulate how to convert paired text and motion into normalized VLM scores (Sec. 4.1). Then, we devise an *entropy-based technique to ensure high-quality VLM score* with low information loss when 3D motion is rendered into video (Sec. 4.2). The process is shown in Fig. 2(b).

### 4.1 FORMULATE VLM SCORES

A text-to-motion model T2M$(\cdot)$ takes a textual prompt $T$ as the input and outputs a motion sequence $M = (m_t)_{t=1}^N$, where $m_t$ is a pose vector at timestep $t$, encoding joint angles, positions, etc. We denote $I_{\text{score}}$ as the instruction template (Fig. 2 (b)) and denote random variable $Y$ as the candidate answer, taking values from $\{y^+ = \text{"yes"}, y^- = \text{"no"}\}$. To compute the alignment score with VLM, we first use Blender software (Community, 2018) to render the motion $M$ into a video $V = (v_t)_{t=1}^N$, using the environment the same as in Section 3.2. A pretrained VLM (e.g., InternVL3-14B(Zhu et al., 2025)) then outputs the likelihood of $y \in Y$ with $I_{\text{score}}$, $T$ and $V$. Finally, we aggregate the likelihood on each $y$ into a normalized distribution representing alignment score:

$$\mathcal{P}_{\text{VLM}}(Y = y^+|I_{\text{score}}, T, V) = \frac{\text{LH}(y^+|I_{\text{score}}, T, V)}{\text{LH}(y^-|I_{\text{score}}, T, V) + \text{LH}(y^+|I_{\text{score}}, T, V)} \quad (1)$$

where LH is conditional likelihood, output by VLM. As examplified in Figure 2 (b), we take *"yes"* as $y^+$, which refers to alignment between text and motion; we take *"no"* as $y^-$, which refers to misalignment between text and motion. Notably, evaluating T2M models with VLMs involves rendering 3D information to 2D information, where accumulated biases and noise (e.g., single-view occlusion) may hamper the quality of VLM scores. To this end, we do not sample the hard prediction (i.e., words) from VLM's continuous output (i.e., likelihood). The likelihood reflects VLM's confidence in the answer and can help us estimate the information loss, we will discuss later.

### 4.2 SELECT LOW ENTROPY SCORES AS EVALUATION

Large language models suffer from the notorious hallucination problem (Huang et al., 2025), and the same is true for VLMs (Liu et al., 2024) — the model may also fabricate answers, even if a definite judgment cannot be drawn from the data input to the model. Fortunately, recent research (Farquhar et al., 2024) has revealed that there is a strong correlation between hallucination and the entropy of the model's output, with speculative hallucinations typically occurring alongside high entropy.

Intuitively, when a rendered video loses important 3D information, such as when the lower body is occluded, we can only guess whether the person in the video is performing a specific leg movement, which is a case of speculative hallucinations. Inspired by the success in speculative hallucination detection, we estimate whether an input rendered video contains sufficient information to answer the textual question by calculating the entropy as follows:

$$H[Y|I_{\text{score}}, T, V] = -\sum_{y \in Y} \mathcal{P}_{\text{VLM}}(Y = y|I_{\text{score}}, T, V) \log\left[\mathcal{P}_{\text{VLM}}(Y = y|I_{\text{score}}, T, V)\right] \quad (2)$$

Based on Eq. (2), we render each motion $M$ into $K$ videos $(V^i)_{i=0}^K$ from different views, and take the final evaluation score $S_{\text{VLM}}$ for each text-motion pair as follows:

$$S_{\text{VLM}}(T, M) = \mathcal{P}_{\text{VLM}}(Y = y^+|I_{\text{score}}, T, V'), \qquad V' = \underset{V \in (V^i)_{i=0}^K}{\arg\min} \; H[Y|I_{\text{score}}, T, V] \quad (3)$$

We validate the entropy-based design in detail in Section 5.4.

## 5 EXPERIMENTS

We first detail the experimental settings (Section 5.1) and baseline metrics for comparison (Section 5.2). Subsequently, we compare the VeMo with existing automatic metrics on our meta-evaluation benchmark (Section 5.3). Finally, we validate our key designs and provide deeper analysis in Section 5.4, and end the Experiments Section with case studies.

## 5.1 Experimental settings

**Implementation details.** We conducted experiments on 1×A100-80G GPU, using LabelStudio (Tkachenko et al., 2020-2025) as frontend for user study, detailed in Appendices. We converted the **Alignment** labels derived from human judgments (see Section 3.2) into binary labels for subsequent meta-evaluation: 0 indicates that the generated motion is not aligned with the prompt used to generate it, while 1 indicates that the generated motion is aligned with its prompt. The meta-evaluation dataset (Sec. 3) is only used for testing, unlike previous works Voas et al. (2023); Wang et al. (2024) where it was also used to fine-tune evaluation models. We take InternVL3-14B (Zhu et al., 2025) as our base in main experiments, and use VeMo as a zero-shot evaluation model, without the use of any text-motion pairs for training or any human label for in-context learning (Dong et al., 2022).

**Datasets and metrics.** We conducted meta-evaluation experiments on the dataset detailed in Section 3, aiming to identify which evaluation scores are most suitable for evaluating text-motion alignment. To this end, we measure the correlation between different evaluation scores and human judgment using the following metrics: 1) **AUC-ROC** measures a binary classifier's ability to distinguish positive from negative classes, defined as the area under the ROC curve (Metz, 1978). ROC curve plots True Positive Rate (TPR) against False Positive Rate (FPR) across all possible decision thresholds. 2) **AUPR** assesses classifier performance (evaluational for imbalanced data) as the area under the Precision-Recall curve (plots Precision vs. Recall). 3) **Kendall's $\tau$** (Kendall, 1945; 1948) measures the correspondence between evaluation scores-based ranking and human scores-based ranking. Values close to 1/-1 indicate strong agreement/disagreement. 4) **Spearman's $\rho$** (Zwillinger & Kokoska, 1999) measures the monotonic association between evaluation scores-based ranking and human scores-based ranking. This one varies between -1 and +1 with 0 implying no correlation. 5) **KS** (Kolmogorov–Smirnov) test statistic (Massey Jr, 1951) quantifies the maximum separation between the cumulative distribution functions of evaluation scores with positive/negative human labels. 6) **Mann-Whitney U Test (p-value)** (McKnight & Najab, 2010) is a nonparametric test. Its null hypothesis is that the positive and negative human-labeled score distributions are identical, and the alternative hypothesis is that the positive human-labeled score distribution is greater.

## 5.2 Baselines.

**L1 Distance** measures the physical distance between each pair of generated and reference motions. However, collecting a comprehensive reference set is very difficult and expensive, limiting the use scenarios (see related work). We adopt the **Minus L1 Distance** to ensure its direction aligns with human labels, since a smaller original L1 Distance corresponds to a positive human label.

**Multimodal Distance (MM Dist)** is computed as the Euclidean distance between the embedding of each pair of generated motion and corresponding text. We use the widely adopted biencoder ((Tevet et al., 2022)) to encode motion and text. We take the **Minus MM Dist** to ensure its direction aligns with human labels, since a smaller original distance corresponds to a positive human label.

**R@K-Precision.** For each generated motion, the text used to generate it and 31 randomly selected mismatched texts in the test set form a prompt pool. This is followed by ranking the MM distances between the motion embedding and the embedding of each prompt in the pool. The MM distance between the motion and its corresponding prompt that ranks top-$K$ is treated as a successful retrieval and the pair is scored as 1; otherwise, scored as 0. We report the results when $K$ takes values from 1 to 3, denoted as **R@1-Precision**, **R@2-Precision**, and **R@3-Precision** respectively.

**MoBERT** (Voas et al., 2023) trains a evaluation model on text-motion data to generate alignment score, denoted as **MoBERT-base**. To further integrate human rating guidance, Voas et al. (2023) tunes the base model on human annotated Faithfulness label and Naturalness label, resulting in **MoBERT-F** and **MoBERT-N** respectively. We also aggregate the two scores with min/max operation. We did not re-implement MoBERT but directly used their open-source model for experiments.

**MotionCritic** (Wang et al., 2024) mainly studies text-independent ranking of motions, and integrates human perception on generated motions to train a evaluation model. Specifically, the evaluation scores indicate whether one motion is judged as superior to another, rather than their distance. We include the officially trained MotionCritic in our baseline without re-implementation.

Table 3: Results of the correlation between different evaluation scores and human judgements, reported on meta-evaluation dataset. ↑ means larger is better, ↓ means lower is better.

| Evaluation Method | AUC-ROC ↑ | AUPR ↑ | KS ↑ | $\tau$ ↑ | $\rho$ ↑ | p-value ↓ |
|---|---|---|---|---|---|---|
| *(reference-based automatic evaluation method)* | | | | | | |
| Minus L1 Distance | 0.627 | 0.628 | 0.204 | 0.180 | 0.220 | <1e-6 |
| *(reference-free automatic evaluation method)* | | | | | | |
| Minus Multimodal Distance | 0.513 | 0.521 | 0.048 | 0.018 | 0.022 | 0.149 |
| R@1-Precision | 0.508 | 0.559 | 0.016 | 0.024 | 0.024 | 0.129 |
| R@2-Precision | 0.504 | 0.573 | 0.007 | 0.009 | 0.009 | 0.343 |
| R@3-Precision | 0.498 | 0.590 | 0.005 | -0.005 | -0.005 | 0.595 |
| MoBERT-base | 0.526 | 0.541 | 0.057 | 0.037 | 0.045 | 0.018 |
| MoBERT-F | 0.532 | 0.548 | 0.087 | 0.045 | 0.055 | 0.005 |
| MoBERT-N | 0.549 | 0.551 | 0.088 | 0.069 | 0.084 | 4e-05 |
| MoBERT-max(F/N) | 0.549 | 0.553 | 0.088 | 0.069 | 0.084 | 4e-05 |
| MoBERT-min(F/N) | 0.534 | 0.549 | 0.086 | 0.048 | 0.058 | 0.003 |
| MotionCritic | 0.506 | 0.517 | 0.027 | 0.009 | 0.010 | 0.312 |
| **VeMo (Ours)** | **0.720** | **0.743** | **0.354** | **0.311** | **0.381** | **<1e-6** |
| User-1 Score | 0.829 | 0.878 | 0.658 | 0.659 | 0.659 | <1e-6 |
| User-2 Score | 0.835 | 0.876 | 0.670 | 0.677 | 0.677 | <1e-6 |
| User-3 Score | 0.833 | 0.877 | 0.665 | 0.666 | 0.666 | <1e-6 |

## 5.3 MAIN RESULTS

Table 3 details the experimental results on meta-evaluation dataset. We compare VeMo with existing evaluation scores, adhering to their official implementations. We mark the **best automatic results in bold** and underline the second-best. **VeMo** significantly outperforms all other automatic methods in all 6 metrics, highlighting a strong correlation with human judgements, even without training on motion data. Among reference-free methods, the maximum improvement of VeMo in the KS statistic is more than 4 times that of the best alternative, and the p-value $< 1e - 6$, indicating that the scores with positive human labels are generally and significantly higher than those with negative human labels; In terms of the correlation coefficients $\tau$ and $\rho$, VeMo also achieves a 4-fold improvement, highlighting a strong correlation between VeMo and human judgments; AUC-ROC and AUPR discuss the performance of evaluation scores used in binary classification across a wider range of thresholds, and VeMo also achieves top-1 performance, with an improvement of up to 0.171. Based on the results, we also make a few comparisons as follows.

First, a lower KS statistic $(0.048)$ indicates that the distribution difference between the **Multimodal Distance** with positive human labels and that with negative human labels is small. The $\tau$ and $\rho$ coefficients between are close to 0, indicating that the Minus Multimodal Distance has a very weak correlation with human labels, and using Multimodal Distance as an evaluation score to assess whether the text and generated motion are aligned is not trustworthy. The standard **R@K-Precision** is calculated based on the Minus Multimodal Distance, so we arrive at the same conclusion.

Second, **MoBERT** is the second-performing reference-free method. MoBERT-base shows a negative correlation with human judgment. After tuning with human feedback, the evaluation scores output by MoBERT-F and MoBERT-N show a positive correlation with human evaluations. Yet, compared to VeMo, which neither incorporates any human feedback on motion nor has been trained on motions, MoBERT still has lower discriminative power in distinguishing positive and negative human labels. We also notice that MotionCritic, a evaluation model that only takes generated motion as input, performs worse than the classic multimodal distance, indicating that it is necessary to consider both the text and the generated motion when evaluating text-motion alignment.

Third, the $\tau$ and $\rho$ coefficients for **users' scores** are close to 0.7, outperforming all other evaluation scores—indicating that no automatic metric can yet fully replace human evaluations. Furthermore, the coefficients of reference-free baselines are less than 0.1, and the $\rho$ coefficient of reference-based **L1 distance** exceed 0.2, indicating that existing reference-free methods not only have a weak correlation with human evaluations but also cannot replace reference-based methods. In contrast, $\tau$ and $\rho$ for VeMo exceed 0.3, demonstrating that our approach not only achieves a meaningful correlation with human evaluations but also can outperform reference-based evaluation methods.

Table 4: Agreement (Krippendorff's Alpha) between users and oracle annotators. User id (U) versus Alignment (A) labels, Faithfulness (F) labels, Naturalness (N) labels.

| U | A | F | N | U | A | F | N | U | A | F | N |
|---|---|---|---|---|---|---|---|---|---|---|---|
| 1 | 0.6566 | 0.6376 | 0.7356 | 2 | 0.6681 | 0.6564 | 0.7896 | 3 | 0.6563 | 0.6348 | 0.7505 |

Table 5: Impact of view selection on the correlation between VeMo scores and human judgments. The results are reported on MDM split, ↑ means larger is better, ↓ means lower is better.

| VeMo (view selection) | AUC-ROC ↑ | AUPR ↑ | KS ↑ | $\tau$ ↑ | $\rho$ ↑ | p-value ↓ |
|---|---|---|---|---|---|---|
| InternVL3-14B (human-opt view) | **0.723** | 0.740 | 0.342 | **0.315** | **0.385** | <1e-6 |
| InternVL3-14B (min entropy view) | 0.720 | **0.743** | **0.354** | 0.311 | 0.381 | <1e-6 |
| InternVL3-14B (random view) | 0.711 | 0.734 | 0.322 | 0.299 | 0.366 | <1e-6 |
| InternVL3-14B (max entropy view) | 0.706 | 0.722 | 0.319 | 0.291 | 0.356 | <1e-6 |

## 5.4 ANALYSIS

**What is the agreement between naive users and oracle annotators?** We present in Table 4 the inter-annotator agreements (IAA) between naive users and oracle annotators, calculated using Krippendorff's Alpha (Krippendorff, 2011). Merely by using the descriptions of the criteria in Section 3.2 to instruct naive users to complete single-choice questions, we can achieve high inter-annotator agreement. This conclusion is consistent with the finding regarding the performance of user scores from the main experiment presented in Table 3. Additionally, we use LabelStudio (Tkachenko et al., 2020-2025) as the frontend for user annotation, see **Appendices** for more details.

**Does entropy-based view selection work?** Table 5 validates entropy-based design (Sec. 4.2) in VeMo. We take the human optimized rendering view (Sec. 3.2) as $V^0$ (*human-opt view*) and randomly rotate the camera around the human body to consider another view $V^1$. We randomly extract a view from $\{V^0, V^1\}$ for evaluation marked *random view*. We select the view with the highest/lowest entropy from $\{V^0, V^1\}$, resulting in the evaluation of *max entropy view* and *min entropy view*. The results of the view corresponding to the lowest entropy of VLM scores are close to those of VLM scores calculated using the human-optimized view, and outperform those using the random view and the max entropy view. This indicates that using low-entropy estimation is a powerful solution for reducing labor costs associated with setting up rendering environments. Additionally, this demonstrates that within our VeMo framework, the predictive entropy of large models can be used to analyze the reliability of input data. Detailed rendering settings can be found in the **Appendices**.

**What is the performance of VeMo when it is based on different models?** Table 9 shows that InternVL3-14B consistently achieves top-1 performance across all metrics. Because the APIs for private VLMs are expensive, difficult to reproduce, and typically inaccessible in terms of per-token likelihood, we only consider the following open-source models for comparison: (1) *Multimodal Models Supporting Video Input* trained on extensive multimodal data (e.g., images, text, tool interactions), which can jointly encode text and video; (2) *Video-Text Foundation Models* focusing on video temporal reasoning, which can also jointly encode text and video; (3) *Video-Text Representation Learning Models*, which encode text and video into vectors independently and finally compute the inner product score. The results show that Video-Text Representation Learning Models exhibit significantly poor performance, with both $\tau$ and $\rho$ coefficients being far below 0.3; In contrast, the coefficients of both *Multimodal Models Supporting Video Input* and *Video-Text Foundation Models* are around 0.3, which demonstrates the importance of jointly encoding text and video.

**Does the number of frames input to the VLM matter?** Table 9 in Appendices shows the results for VeMo using different numbers of frames uniformly sampled from input video. In our experimental setup, exceeding 32 frames results in Out of Memory for InternVL3-14B (Zhu et al., 2025) and InternVL3.5-14B (Wang et al., 2025a); the InternVideo2.5 (Wang et al., 2025b) model has only 8B parameters and supports an input of 128 frames, while ViCLIP-L-14 (Wang et al., 2023) only supports an input of 8 frames. Overall, using more frames leads to a slight improvement in performance. However, when 128 frames are input, InternVideo2.5 exhibits a slight performance drop, indicating the presence of saturation. Nevertheless, the magnitude of these performance changes is relatively small and does not affect the conclusions drawn from the main experiment.

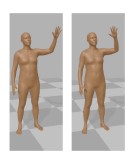
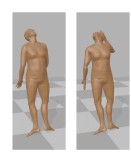

| | | |
|---|---|---|
| T: "a person waves a friendly hello." | "multimodal dist": 11.135 (smaller → better match) | |
| "human label": 1 (larger → better match) | "MoBERT": 0.373 (larger → better match) | |
| "VeMo (Ours)": 0.999 (larger → better match) | "MotionCritic": -11.257 (larger → better match) | |

| | | |
|---|---|---|
| T: "a person opens and drinks from a container." | "multimodal dist": 8.609 (smaller → better match) | |
| "human label": 0 (larger → better match) | "MoBERT": 0.418 (larger → better match) | |
| "VeMo (Ours)": 0.013 (larger → better match) | "MotionCritic": -7.034 (larger → better match) | |

Figure 3: Two interesting cases where the evaluation scores from VeMo clearly align with human labels, while those from others are ambiguous. Prompts (T) are taken from the HumanML3D testset.

Table 6: Extending VeMo score to HumanML3D benchmark for T2M model evaluation.

| T2M Model | VeMo ↑ | FID ↓ | MM Dist ↓ | R@1 ↑ | R@2 ↑ | R@3 ↑ |
|---|---|---|---|---|---|---|
| MDM (Tevet et al., 2022) | .6171±.0024 | .544±.044 | 5.566±.027 | .491±.001 | .681±.001 | .782±.001 |
| MotionGPT (Jiang et al., 2023) | .5723±.0034 | .232±.008 | 3.096±.008 | .492±.003 | .681±.003 | .778±.002 |
| StableMofus. (Huang et al., 2024) | .6528±.0001 | .098±.003 | 2.770±.006 | .553±.003 | .748±.002 | .841±.002 |
| MLD-M (Dai et al.) | .6626±.0002 | .073±.003 | 2.810±.008 | .548±.003 | .738±.003 | .829±.002 |
| MotionLCM-V2 (Dai et al.) | .6638±.0005 | .072±.003 | 2.767±.007 | .546±.003 | .743±.002 | .837±.002 |
| Real | .6825±.0000 | .002±.000 | 2.974±.008 | .511±.003 | .703±.003 | .797±.002 |

**Benchmark of T2M Approaches.** We base VeMo on InternVL3-14B (32-frame) with human-opt view selection and report VeMo scores on the HumanML3D benchmark covering representative T2M models to give the community a clear baseline for comparison. As shown in Table 6, we observed that several T2M models outperform the ground truth on certain reference-free metrics (e.g., MM Dist and R@K-Precision), which suggests those evaluators can be overfit or "hacked." FID is a reference-based method and has a high correlation with the VeMo metric.

**Impact of K (number of views).** Detailed experiments and analyses can be found in Appendices (Table 12, Table 13), and we directly present the important conclusions: (1) Selecting the min-entropy view (i.e., the most confident view) consistently outperforms selecting the max-entropy view when $K > 1$. (2) The largest gain occurs when moving from $K = 1$ to $K = 2$; beyond $K = 2$ performance quickly saturates. Thus $K = 2$ provides a strong balance between reliability and cost.

**More in-depth analyses.** To further study the performance boundaries of VeMo, we provide extra results in Appendices, including **Computational Overhead, Efficiency Tradeoff, Acceleration, Stability of the VLM-Generated Scores, Future Direction and etc.**

**Case studies.** Figure 3 shows interesting cases where the evaluation scores from VeMo clearly align with human labels, while those from other methods are ambiguous. In the left case, the prompt contains abstract concepts, requiring an understanding that people usually wave hand to express a friendly "hello." VeMo **faithfully** grasps the underlying action corresponding to the prompt, assigning high confidence score that align with human label; while baseline methods fail to comprehend the complex semantics in the sentence. The person in the right case drinks water with the back of their head **unnaturally**. VeMo recognizes that "anti-human style" of the human motion, thus assigning a low score and determining that the generated motion is unaligned with the prompt. Taking the two cases into consideration, we find that VeMo can understand the complex semantics and human style while evaluating generated motions. This further validates the effectiveness of VeMo.

# 6 CONCLUSION

We considered the evaluation of T2M alignment, proposed a new meta-evaluation benchmark to solve the problem that there is no shared testbed to fairly compare the generalizability of automatic evaluators. Moreover, we use the VLM to solve T2M evaluation, and devise an entropy-based technique to foster a high-quality VLM score when 3D motion is rendered into 2D video. Our method, named VeMo benefits T2M evaluation from internet-scale text-vision data and achieves human-aligned evaluation performance. This evaluation method can potentially not only provide a fairer comparison for different T2M models but also offer more accurate feedback for the development of new models. *Extra analyses and codes can be found in the Appendices and Supplemental.*

STATEMENT

**Ethics statement.** The evaluation method we propose does not raise new ethical concerns, but may inherit the internal biases of the video language model on which VeMo is based. Key biases include Western-centric cultural representation imbalance, implicit cultural stereotypes, and limited grasp of contextual cultural nuance—all of which could skew VeMo's assessments of human movements, gestures, or social interactions across diverse cultural contexts, compromising the framework's fairness and generalizability. Please refer to (Nayak et al., 2024) for more details.

**Reproducibility statement.** We provide in the Experimental Section and Appendices a clear setup for reproducibility. We also upload the code of the full evaluation pipeline, as well as the resources of the main experiment as supplemental materials. This ensures reproducibility.

**Use of LLMs in writing.** We only use LLMs to polish writing, e.g., grammar/spelling checking. We also double-check the polished texts to try our best to optimize the readers' experience.

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

# A APPENDICES

## A.1 DETAILS ON BENCHMARK DATA COLLECTION

**Data source.** We based our benchmark data on HumanML3D (Guo et al., 2022), a recent dataset. HumanML3D textually re-annotating motion capture from the AMASS (Mahmood et al., 2019) and HumanAct12 (Guo et al., 2020) collections. It contains 14,616 motions annotated by 44,970 textual descriptions, split in train, val, test sets. The train and val splits of HumanML3D are widely adopted to train T2M models. We take the descriptions from the HumanML3D's test set as prompts to generate and evaluate motions. To ensure the diversity and representativeness of the prompts, we use Sentence Transformer, i.e., all-mpnet-base-v2 (Song et al., 2020), to encode all prompts for deduplication. Specifically, we recursively merged pairs of clusters of sample data, using the cosine distance given by these embeddings, with a clustering threshold of 0.8, resulting in approximately 1.5k prompts. Finally, we customized more conditions to further filter the remaining 1.5k prompts. The removal conditions include spelling errors in the action descriptors in the prompts, or prompts describing dexterous hand movements, gaze, and other actions that do not belong to the HumanML3D joints. In the end, 1101 texts remained.

**Motion generation.** A trained T2M model will take textual motion annotations as input and output motion sequence $M = (m_t)_{t=1}^N$ of human poses represented by joint rotations or positions $m_t \in \mathcal{R}^{J \times D}$. $J$ is the number of joints and $D$ is the dimension of the joint representation. Specifically, we employ a diffusion-based MDM (Tevet et al., 2022) and an autoregressive MotionGPT (Jiang et al., 2023) to generate motion data from 1101 selected prompts for subsequent meta-evaluation. Because the codebases of these two models are widely adopted as the foundation for other methods Tevet et al. (2024); Han et al. (2025), and both models are trained on the HumanML3D's trainset and support the animation of body actions for the 22-joint SMPL human model. For MDM, we use the official checkpoint "humanml_trans_dec_512_bert-50steps"; for MotionGPT, we also use the official checkpoint "OpenMotionLab/MotionGPT-base". Finally, we obtain generated motions from MDM and MotionGPT, and there are a total of 2202 text-motion pairs.

## A.2 DETAILS ON VISUALIZATION SETTINGS

To annotate a generated motion sequence $M$ and the text used for generation, we first use Blender (Community, 2018) software to convert the generated motion $M$ into skinned human model video $V$. Specifically, we first use the smplx Python package to create a neutral SMPL model (Loper et al., 2023). Then, we run SMPLify (Bogo et al., 2016) to convert the motion sequence M into a 3D voxel representation (.obj file), which can be directly imported into the Blender environment. Finally, we use Blender 4.0.2 to render the voxels and generate the videol as follows.

We configure the rendering resolution to 1088×1088 pixels, with PNG set as the output format for intermediate frame images to ensure high-quality image data for subsequent video compilation. The scene background is configured as a natural white color (RGB: 1.0, 1.0, 1.0) with an intensity value of 0.6, which avoids overexposure while ensuring the human model stands out clearly against the background. A chessboard-patterned floor is added to the scene to provide spatial reference and enhance visual layering. This floor has a size of 10×10 units and is divided into 10×10 grid divisions; its vertical position (Z-axis) is aligned with the lowest point of the human model's bounding box to ensure it fits naturally under the model. The floor uses a semi-transparent material (transparency set to 0.5) based on the Principled BSDF shader, and a chessboard texture is applied via UV smart projection to ensure the pattern is evenly distributed and displayed correctly. Additionally, the floor is set to be unselectable to prevent accidental modification during the rendering process. For the human model, a custom "DarkBronzeSkinMaterial" is developed to simulate a realistic skin-like appearance. The material's base color is set to an RGB value of (0.3, 0.15, 0.07) (a deep brown with warm bronze undertones), the metallic attribute is adjusted to 0.4 to enhance subtle reflective properties, and the roughness is set to 0.6 to soften excessive gloss, resulting in a natural texture that better showcases the model's contour details and motion changes.

Lighting is provided by a single SUN-type light source with an energy value of 4.5 to ensure sufficient and uniform illumination of the human model. The light source is fixed at the spatial position (-4, -6, 6) and rotated to face the geometric center of the human model—this rotation is calculated by converting the vector from the light source to the model's center into an Euler angle, ensuring the

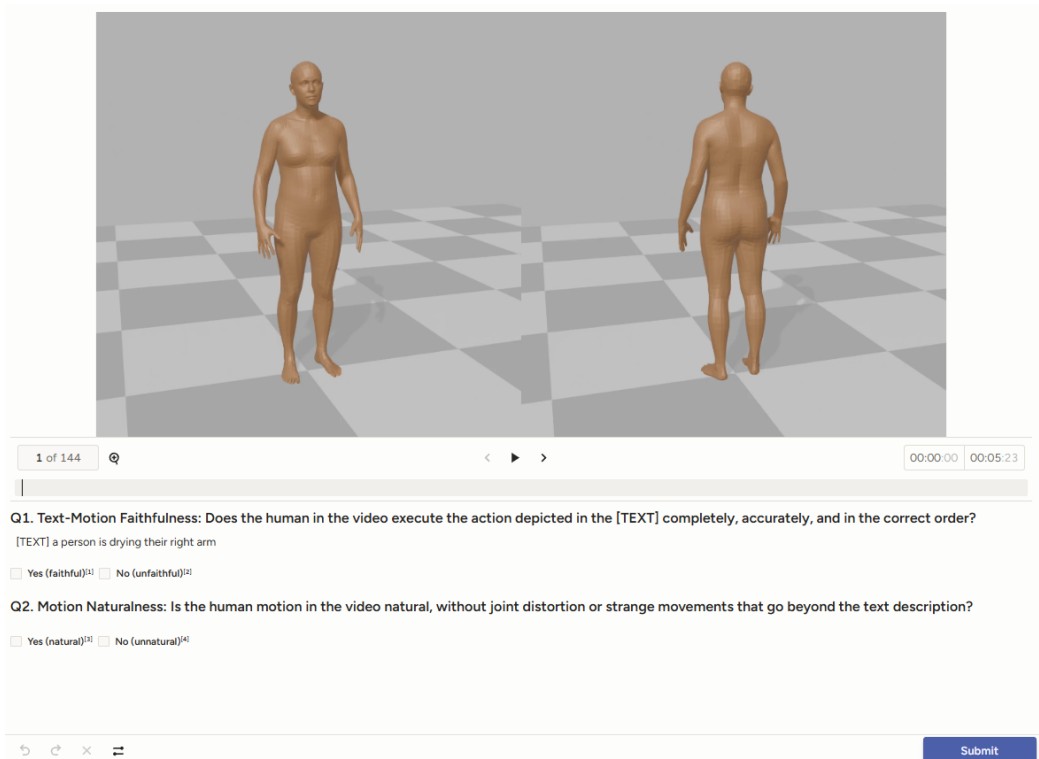

Figure 4: The user-interface of LabelStudio used for human annotation.

Table 7: Model-level correlation between different evaluation scores and human judgements, reported on meta-evaluation dataset. ↑ means larger is better.

| Evaluation Method | $\tau$ ↑ | $\rho$ ↑ | Evaluation Method | $\tau$ ↑ | $\rho$ ↑ |
|---|---|---|---|---|---|
| *(reference-based automatic evaluation method)* | | | | | |
| Minus L1 Distance (w/ ref.) | 0.6138 | 0.8172 | Minus FID (w/ ref.) | 0.6032 | 0.8066 |
| *(reference-free automatic evaluation method)* | | | | | |
| Minus Multimodal Distance | 0.5503 | 0.7600 | MoBERT-base | 0.4762 | 0.7043 |
| R@1-Precision | 0.5745 | 0.7922 | MoBERT-F | -0.4021 | -0.5779 |
| R@2-Precision | 0.5676 | 0.7595 | MoBERT-N | -0.4127 | -0.6622 |
| R@3-Precision | 0.6096 | 0.7988 | MoBERT-max(F/N) | -0.4868 | -0.7193 |
| MotionCritic | 0.5609 | 0.7901 | MoBERT-min(F/N) | -0.4339 | -0.6110 |
| **VeMo (min entropy view)** | 0.7196 | 0.8774 | **VeMo (human-opt view)** | 0.7090 | 0.8834 |
| User-1 Score | 0.6915 | 0.8758 | User-2 Score | 0.5532 | 0.7726 |

light rays are directed toward the model and minimizing harsh shadows that could obscure motion details. For camera configuration, a new camera is created for each frame of the motion sequence to maintain consistent framing of the human model. The camera's position is determined by offsetting the human model's geometric center by a fixed vector (-1, -3, 0.6); similarly to the light source, the camera is rotated to face the model's center (using the vector from the camera to the model's center converted to an Euler angle). This setup ensures that the human model's entire body remains fully visible in each frame, and the model occupies more than 1/10 of the frame area as required. After rendering all motion frames into individual PNG images, the images are compiled into a video file using the H.264 codec (libx264) with a frame rate of 20 FPS and a quality parameter of 9. This codec and parameter combination balances video quality and file size, resulting in a smooth motion video that clearly presents the details of the generated motion sequence. Finally, we use LabelStudio as a frontend for human annotation, and the user-interface is shown in Figure 4.

Table 8: Extra ablation result on usefulness of Eq. (1).

| VeMo, human-opt view | AUC-ROC ↑ | AUPR ↑ | KS ↑ | $\tau$ ↑ | $\rho$ ↑ |
|---|---|---|---|---|---|
| InternVL3-14B (w/ Eq. (1)) | **0.723±.000** | 0.740±.000 | **0.342±.000** | **0.315±.000** | **0.385±.000** |
| InternVL3-14B (w/o Eq. (1)) | 0.627±.006 | **0.743±.004** | 0.254±.012 | 0.262±.013 | 0.262±.013 |

## A.3 EXPERIMENTS ON MODEL-LEVEL CORRELATION

**Settings.** For the data belonging to the MDM split, we randomly divide it into two groups and repeat this process 10 times, resulting in a total of 20 random sub-splits. We perform the same procedure for the MotionGPT split, which also generates 20 random sub-splits. Ultimately, we obtain 40 sub-splits in total, with each sub-split containing approximately 500 data samples. Subsequently, each sub-split is analyzed independently: first, we calculate the FID score following (Tevet et al., 2022); then, we sum up all text-motion scores for each evaluation method. Finally, for each metric, we have a sequence of model-level scores with a length of 40, which can be used to compute the correlation coefficients $\tau$ and $\rho$. Table 7 reports model-level correlation of different evaluation methods.

**Based on Table 7, we have several findings:** 1) First, we observe that the model-level MoBERT score of the model fine-tuned on human-labeled samples exhibits a negative correlation with human scores, which is contrary to the case at the sample level (see Table 3). This may be attributed to the insufficient amount of human-labeled data, which causes the fine-tuned model to suffer from overfitting and assign extreme scores for false positive/negative samples. 2) Second, among automatic methods, VeMo ranks first in terms of model-level correlation, followed by reference-based methods. This is consistent with the findings from the sample-level evaluation (Table 3), indicating that both VeMo and reference-based methods can provide stable and consistent evaluations. 3) Finally, VeMo's model-level scores achieve or even surpass human user scores. The reason for this is that users are instructed to make binary selections, and inconsistent cases largely affect model-level scores. In contrast, VeMo can generate intermediate scores for cases it is uncertain about, thereby improving its model-level performance. Overall, although model-level scores cannot be used to evaluate text-motion alignment, the experimental results demonstrate that VeMo has the potential to evaluate the overall performance of T2M models, which we leave for future research.

## A.4 EXTRA ANALYSIS

**What about using VLM's output sentence instead of the predicted distribution?** To study the usefulness of Eq. (1), we replace Eq. (1) with matches of "yes" or "no" from the sentences generated by the VLM. This produces binary (i.e., 0/1) prediction scores, then we compare these scores with our original distribution-based scores. Note that the entropy of the output sentence of the VLM cannot be calculated, so we evaluate it under the same setting of human-opt view as in Table 5. The results in Table 8 show that VeMo w/ Eq. (1) outperforms VeMo w/o Eq. (1) in terms of AUC, KS, $\tau$ and $\rho$ (most important), indicating that Eq. (1) can significantly enhance the correlation between the evaluation scores and humans. VeMo w/o Eq. (1) yields binary scores, and the precision and recall under most thresholds are the same, which is slightly beneficial for AUPR calculation. Notably, the VeMo scoring procedure (soft distributional scoring per Eq. (1)) yields stable, zero-variance scores for deterministic input video. The VeMo scores are substantially more reliable than scores obtained via naive deterministic decoding of VLM outputs (i.e., w/o Eq. (1)).

**Computational overhead and efficiency tradeoff of the VeMo pipeline.** Our full VeMo pipeline consists of three stages: Joint-to-Mesh, Mesh-to-Video, and VLM inference. Converting a rendered 3D mesh into multi-view videos does not increase the Joint-to-Mesh overhead. For the rendering process, we measured the time and peak RAM using the Rendering-to-Body-Model video converter: the per-motion time and peak RAM for the Joint-to-Mesh stage are 182s and 793MiB, respectively, while the corresponding values for the Mesh-to-Video stage are 31s and 256MiB. The runtime, memory footprint, and evaluation performance of VeMo under different VLM configurations (human-opt view) are shown in Table 10. The primary time bottleneck is rendering, but rendering is highly parallelizable because its peak RAM is low. The primary RAM bottleneck is VLM inference; using smaller VLMs (for example, InternVL3-1B) reduces memory requirements at the cost of modest performance degradation. This trade-off makes VeMo practical for different resource budgets.

Table 9: Ablation studies on number of input frames and VLM used as VeMo.

| VeMo (num frames) | AUC-ROC ↑ | AUPR ↑ | KS ↑ | $\tau$ ↑ | $\rho$ ↑ | p-value ↓ |
|---|---|---|---|---|---|---|
| *(Multimodal Models Supporting Video Input)* | | | | | | |
| InternVL3-14B (32 frames) | 0.723 | 0.740 | 0.342 | 0.315 | 0.385 | <1e-6 |
| InternVL3-14B (8 frames) | 0.720 | 0.738 | 0.343 | 0.311 | 0.381 | <1e-6 |
| InternVL3.5-14B (32 frames) | 0.709 | 0.734 | 0.331 | 0.296 | 0.363 | <1e-6 |
| InternVL3.5-14B (8 frames) | 0.698 | 0.721 | 0.306 | 0.281 | 0.343 | <1e-6 |
| *(Video-Text Foundation Models)* | | | | | | |
| InternVideo2.5 (128 frames) | 0.684 | 0.700 | 0.292 | 0.260 | 0.318 | <1e-6 |
| InternVideo2.5 (32 frames) | 0.688 | 0.706 | 0.292 | 0.266 | 0.325 | <1e-6 |
| InternVideo2.5 (8 frames) | 0.669 | 0.688 | 0.263 | 0.239 | 0.292 | <1e-6 |
| *(Video-Text Representation Learning Model)* | | | | | | |
| ViCLIP-L-14 (8 frames) | 0.559 | 0.543 | 0.099 | 0.084 | 0.103 | 1e-06 |

Table 10: VeMo Performance Under Different VLM Configurations (Human-Opt View).

| VeMo (human-opt view) | Per-Video Time/Peak-RAM | AUC-ROC | AUPR | KS | $\tau$ | $\rho$ |
|---|---|---|---|---|---|---|
| InternVL3-14B (32-frame) | 1.597s / 33221MiB | 0.723 | 0.740 | 0.342 | 0.315 | 0.385 |
| InternVL3-14B (8-frame) | 0.415s / 30000MiB | 0.720 | 0.738 | 0.343 | 0.311 | 0.381 |
| InternVL3-8B (32-frame) | 0.889s / 18332MiB | 0.687 | 0.706 | 0.291 | 0.264 | 0.324 |
| InternVL3-8B (8-frame) | 0.233s / 15998MiB | 0.683 | 0.701 | 0.284 | 0.258 | 0.316 |
| InternVL3-1B (32-frame) | 0.295s / 4470MiB | 0.630 | 0.627 | 0.215 | 0.184 | 0.225 |
| InternVL3-1B (8-frame) | 0.084s / 2503MiB | 0.642 | 0.645 | 0.231 | 0.201 | 0.246 |

**Efficient version of VeMo.** Distilling VLM into a smaller scoring model is attractive for efficiency. However, T2M generation exhibits many diverse, valid solutions and limited coverage of motion space; a distilled model risks overfitting to limited T2M data in much the same way as current reference-free evaluators. To address the practical time bottleneck from mesh rendering, we explored a lightweight alternative: directly visualizing joint trajectories as stick-figure videos (i.e., skipping Joint-to-Mesh). This eliminates the rendering overhead while preserving temporal joint information for the VLM. The runtime and memory profile for this converter is negligible: the Visualizing-to-Stick-Figure-Video converter achieves a per-frame time of 0.007s with a peak RAM of 0MiB. As shown in Table 11, we evaluated VeMo using stick-figure videos. Although absolute performance drops relative to full-body renderings, VeMo on stick figures still substantially outperforms the best reference-free baseline and is therefore suitable for rapid, online analyses and iterative workflows.

Table 11: VeMo performance on stick-figure videos.

| VeMo on Stick-Figure-Video | AUC-ROC | AUPR | KS | $\tau$ | $\rho$ |
|---|---|---|---|---|---|
| InternVL3-14B (32-frame) | 0.608 | 0.626 | 0.145 | 0.153 | 0.187 |
| InternVL3-14B (8-frame) | 0.619 | 0.633 | 0.176 | 0.169 | 0.206 |
| InternVL3-8B (32-frame) | 0.604 | 0.607 | 0.148 | 0.147 | 0.180 |
| InternVL3-8B (8-frame) | 0.606 | 0.608 | 0.159 | 0.150 | 0.184 |
| InternVL3-1B (32-frame) | 0.560 | 0.548 | 0.108 | 0.085 | 0.104 |
| InternVL3-1B (8-frame) | 0.571 | 0.563 | 0.118 | 0.100 | 0.123 |

**Impact of K (the number of views) on evaluation performance.** We pre-extract six views per motion by uniformly rotating the camera around the body (the engineered "human-opt" view plus five random rotations). For evaluation we sample K views without replacement from these six, and report metrics for two view-selection strategies: the minimum-entropy view (Table 12) and the maximum-entropy view (Table 13). Here is the Conclusion:

- Selecting the minimum-entropy view (i.e., the most confident view) consistently outperforms selecting the maximum-entropy view when $K > 1$.

- The largest gain occurs when moving from $K = 1$ to $K = 2$; beyond $K = 2$ performance quickly saturates. Thus $K = 2$ provides a strong balance between reliability and cost.
- Generating additional views incurs roughly 30 seconds of rendering per extra perspective per motion. Given this cost, we recommend using the engineered human-opt view (i.e., human-opt, $K = 1$), which already achieves performance close to a higher-$K$ regime.

Table 12: Min-entropy based view selection.

| $K$ | AUC-ROC | AUPR | KS | $\tau$ | $\rho$ |
|---|---|---|---|---|---|
| 1 | 0.706 | 0.729 | 0.304 | 0.291 | 0.357 |
| 2 | 0.721 | 0.742 | 0.346 | 0.312 | 0.382 |
| 3 | 0.719 | 0.742 | 0.349 | 0.310 | 0.380 |
| 4 | 0.721 | 0.746 | 0.356 | 0.313 | 0.384 |
| 5 | 0.723 | 0.748 | 0.361 | 0.315 | 0.386 |
| 6 | 0.721 | 0.745 | 0.358 | 0.312 | 0.382 |

Table 13: Max-entropy based view selection.

| $K$ | AUC-ROC | AUPR | KS | $\tau$ | $\rho$ |
|---|---|---|---|---|---|
| 1 | 0.706 | 0.729 | 0.304 | 0.291 | 0.357 |
| 2 | 0.711 | 0.728 | 0.324 | 0.298 | 0.365 |
| 3 | 0.705 | 0.720 | 0.312 | 0.290 | 0.355 |
| 4 | 0.701 | 0.718 | 0.308 | 0.284 | 0.347 |
| 5 | 0.699 | 0.715 | 0.308 | 0.282 | 0.345 |
| 6 | 0.699 | 0.712 | 0.308 | 0.281 | 0.344 |

**Combining information from multiple views.** To probe whether multi-view fusion helps, we ran an experiment using videos where each frame contains two synchronized views (one human-optimal view and one randomly rotated view). Evaluation with InternVL3-14B (32-frame) yields the following metrics: For the video containing two synchronized views, the performance metrics are AUC-ROC of 0.716, AUPR of 0.740, KS of 0.348, Kendall's tau ($\tau$) of 0.306, and Spearman's rho ($\rho$) of 0.374. In contrast, for the video using only the human-opt view, the corresponding metrics are AUC-ROC of 0.723, AUPR of 0.740, KS of 0.342, $\tau$ of 0.315, and $\rho$ of 0.385. We find that naively combining multiple views can confuse current VLMs and slightly degrade VeMo performance. This suggests that effective multi-view integration likely requires improvements in VLM video understanding (better temporal and multi-view fusion). We therefore identify multi-view fusion as an important avenue for future work rather than claiming a simple aggregation rule is sufficient today.

A.5 FUTURE DIRECTION.

**Evaluation on the extent of failure cases.** The VeMo score was intentionally designed to answer the most fundamental and objectively measurable question in text-to-motion (T2M): Does the generated motion match the text? Current automatic metrics often struggle to reliably distinguish even coarse-grained "yes/no" alignment, so we focused first on this basic and well-defined problem. Fine-grained degrees of adherence (e.g., mostly correct with small mistakes vs. completely wrong) are important; however, they require richer annotations and more capable evaluation tools. We view fine-grained alignment as an important direction for future work.

**Scaling the meta-evaluation.** The meta-evaluation benchmark introduced in this work has a focused goal: to construct a fair human-rating reference and to verify the basic evaluation effectiveness of T2M evaluators. Our experiments and analyses are framed around this objective, and they demonstrate that the VLM-based T2M score is substantially closer to human ratings than existing automatic evaluators, which is an important step toward improving T2M evaluation. Notably, scaling the meta-evaluation to cover a much wider range of motion complexity and more T2M models is valuable in future but also labor-intensive and outside the core contributions claimed in paper.

**Boundaries of entropy-based view selection.** VLMs may give incorrect high-entropy predictions for input videos, these edge cases are usually considered out-of-distribution for the model (Liu et al., 2024; Farquhar et al., 2024). This limitation reflects a gap between VLM video-understanding capabilities and human perception. Addressing these corner cases will require VLMs with more human-like motion perception and higher-fidelity grounding. Finally, perfect agreement between any automatic evaluator and human raters is unattainable (Table 3), since T2M alignment is subjective and multimodal. Our primary objective is to narrow the gap between automatic T2M evaluators and human judgment; tackling remaining edge cases is an important direction for future work.

**Beyond the offline analysis presented**, we believe VeMo can provide concrete value for T2M training in two practical ways: (1) Training-data curation. Recent efforts augment training sets with motions recovered from video (Ding et al., 2025). VeMo can serve as an automatic filter to remove low-quality or noisy converted motions in trainset. (2) Training-time reward shaping. VeMo can be

combined with offline reinforcement-learning schemes as an auxiliary reward signal; by periodically re-scoring and updating offline data, models can be steered toward higher-quality generations.

### A.6 LIMITATIONS AND BROADER DISCUSSION

Our research focuses on using video-language model (VLM) as evaluator but shares VLM limitations. To be more clear, this evaluation score is influenced by inherent biases in VLMs, as identified in studies by Fei et al. (2023). Addressing this requires strategies for fair, interpretable outcomes from complex models, presenting a promising research area. We believe that improvements in automatic evaluation metrics can be used to generate supervisory signals to guide the performance of T2M models to reach the cognitive level of VLMs. Some recent works have attempted to train T2M models using generated answers from VLMs indirectly (Han et al., 2025; Pappa et al., 2024), but they still rely on old metrics such as FID, Multimodal distance for evaluation. Our work not only provides support for these empirical usages, but also studies the loss associated with the scores generated by VLMs and reveals the gap with human-level performance.

