# OpenReview forum: "Video Language Models are Human-Aligned Evaluators for Text to Motion Generation"
_ICLR.cc/2026/Conference — ICLR 2026 Conference Withdrawn Submission_

### Official Review · Reviewer_4br3 · 2025-10-27

**Soundness:** 3
**Presentation:** 3
**Contribution:** 2
**Rating:** 4
**Confidence:** 4

**Summary:**

This paper introduces VeMo, a method for evaluating text-to-motion (T2M) generation by leveraging Video Language Models (VLMs). The core idea is to render 3D generated motions into 2D videos and use a VLM to assess the alignment between the video and the input text prompt. To mitigate information loss from the 3D-to-2D projection, the method employs an entropy-based view selection technique to identify the most informative camera angle for evaluation. Furthermore, the authors contribute a new meta-evaluation benchmark, including human annotations for text-motion alignment, to facilitate a standardized comparison of different evaluation metrics. Experimental results demonstrate that VeMo's scores correlate more strongly with human judgments than existing reference-based and reference-free methods.

**Strengths:**

1. Important Problem: The paper addresses a critical bottleneck in the T2M field. Developing reliable, automatic evaluation metrics that align with human perception is essential.
2. Intuitive and Effective Method: The core idea of rendering 3D motions into 2D videos to leverage the powerful semantic understanding of pre-trained VLMs is clever and well-motivated. As demonstrated in the main results (Table 3), this approach proves effective, achieving a significantly higher correlation with human judgments across multiple metrics compared to previous automatic evaluation techniques.
3. Valuable Benchmark: The creation of a meta-evaluation benchmark with coarse- and fine-grained human annotations is a strong contribution to the research community. This resource enables a fair and standardized comparison of existing and future T2M evaluation methods, which has been a significant gap in the field. The quality of the benchmark is supported by a high inter-annotator agreement.

**Weaknesses:**

1. (Major Concern) Computational Overhead: The primary limitation of VeMo is its significant computational expense. The pipeline requires rendering motions into videos, potentially from multiple viewpoints, and then performing inference with a large-scale VLM. This high overhead makes the metric impractical for use as an online reward signal during model training. And for large-scale evaluations, this effectively limits its utility to offline analysis. The paper would be strengthened by a discussion of this limitation and an exploration of potential mitigation strategies, such as distilling the VLM evaluator into a smaller, more efficient scoring model.
2. Lack of Analysis on Number of Views: The paper proposes rendering motion from K different views and selecting the one with the lowest entropy. However, the experiments only compare a human-optimized view against one other randomly generated view. There is no ablation study or discussion on the impact of the number of views (K) on evaluation performance and the trade-off with computational cost. It is unclear how many views are necessary to achieve reliable scores.
3. VLM Score Stability: The paper does not address the potential instability of VLM-generated scores. It is known that even with deterministic decoding settings (e.g., temperature=0), language models can exhibit slight variations in output likelihoods. Since the VLM score is the fundamental output of the metric, an analysis of its variance across multiple runs is necessary to establish the reliability and consistency of VeMo as an evaluator.

**Questions:**

1. Could the authors provide a more detailed discussion on the computational overhead of the VeMo pipeline? Given the expense of rendering and VLM inference, how do they see its practical utility beyond offline analysis, for example, as a reward signal during training?
2. Following up on the computational cost, have the authors explored any mitigation strategies to create a more efficient version of the metric, such as distilling the VLM evaluator into a smaller, specialized scoring model?
3. The paper proposes selecting from K views based on entropy, but the experiments only seem to compare a human-optimized view against one random view. Could the authors provide an ablation study on the impact of K (the number of views) on evaluation performance? What is the trade-off between the number of views, score reliability, and computational cost?
4. What is the stability or variance of the VLM-generated scores? Could the authors provide an analysis of VeMo's reliability across multiple runs on the same inputs (even with deterministic decoding) to establish the consistency of the metric?

---

> ### Author Response · Authors · 2025-11-21
> **Part-1**
>
> Thank you very much for your careful reading and constructive feedback, we appreciate your recognition of our work and the thoughtful suggestions that helped us improve the paper. Below, we provide point-by-point responses to your questions.
>
> ### **Q1, W1. Computational Overhead and Efficiency Tradeoff of the VeMo Pipeline**
> >To find the detailed results and analyses, we refer the reviewer to the official comment posted at the top. Beyond the offline analysis presented there, we believe VeMo can provide concrete value during model training in two practical ways:
> >- Training-data curation. Some recent efforts augment training sets with motions recovered from video. VeMo can serve as an automatic filter to remove low-quality or noisy converted motions before they enter the training pool.
> >- Training-time reward shaping. VeMo can be combined with offline reinforcement-learning schemes as an auxiliary reward signal; by periodically re-scoring and updating offline data, models can be steered toward higher-quality generations.
>
> >In the official comment at the top we also discuss concrete efficiency improvements for reward generation, specifically, strategies to reduce RAM usage and to exploit high parallelism during rendering.
>
>
>
> ### **Q2. Efficient Version of VeMo Scores**
> >We agree that distilling a large VLM into a smaller scoring model is attractive for efficiency. However, T2M generation exhibits many diverse, valid solutions and limited coverage of motion space; a distilled model risks overfitting to limited T2M data in much the same way as current reference-free evaluators.
>
> >To address the practical time bottleneck from mesh rendering, we explored a lightweight alternative: directly visualizing joint trajectories as stick-figure videos (i.e., skipping Joint-to-Mesh). This eliminates the rendering overhead while preserving temporal joint information for the VLM. The runtime and memory profile for this converter is negligible:
> | Video Converter | Per-frame Time/Peak-RAM (Joint-to-Stick-Figure-Video) |
> |---|---|
> | Visualizing-to-Stick-Figure-Video | 0.007s / 0MiB |
>
> > We evaluated VeMo using stick-figure videos. Although absolute performance drops relative to full-body renderings, VeMo on stick figures still substantially outperforms the best reference-free baseline:
> | VeMo on Stick-Figure-Video | AUC-ROC | AUPR | KS | τ | ρ |
> |---|---|---|---|---|---|
> | InternVL3-14B (32-frame) | 0.608 | 0.626 | 0.145 | 0.153 | 0.187 |
> | InternVL3-14B (8-frame) | 0.619 | 0.633 | 0.176 | 0.169 | 0.206 |
> | InternVL3-8B (32-frame) | 0.604 | 0.607 | 0.148 | 0.147 | 0.180 |
> | InternVL3-8B (8-frame) | 0.606 | 0.608 | 0.159 | 0.150 | 0.184 |
> | InternVL3-1B (32-frame)  | 0.560 | 0.548 | 0.108 | 0.085 | 0.104 |
> | InternVL3-1B (8-frame)  | 0.571 | 0.563 | 0.118 | 0.100 | 0.123 |
>
> > Best reference-free baseline
> |  | AUC-ROC | AUPR | KS | τ | ρ |
> |---|---|---|---|---|---|
> | MoBERT | 0.549 | 0.553 | 0.088 | 0.069 | 0.084 |
>
> >**Conclusion.** Stick-figure videos effectively remove the rendering time bottleneck. While there is a measurable drop in absolute evaluation performance, the stick-figure variant still surpasses strong reference-free baselines and is therefore suitable for rapid, online analyses and iterative workflows.

---

> ### Author Response · Authors · 2025-11-21
> **Part-2**
>
> ### **Q3, W2. Impact of K (the number of views) on Evaluation Performance**
> >We pre-extract six views per motion by uniformly rotating the camera around the body (the engineered “human-opt” view plus five random rotations). For evaluation we sample K views without replacement from these six, and report metrics for two view-selection strategies: the minimum-entropy view and the maximum-entropy view.
>
> >**Minimum entropy based view selection**
> | K |  AUC-ROC | AUPR | KS | τ | ρ |
> |---|---|---|---|---|---|
> | 1 | 0.706 | 0.729 | 0.304 | 0.291 | 0.357 |
> | 2 | 0.721 | 0.742 | 0.346 | 0.312 | 0.382 |
> | 3 | 0.719 | 0.742 | 0.349 | 0.310 | 0.380 |
> | 4 | 0.721 | 0.746 | 0.356 | 0.313 | 0.384 |
> | 5 | 0.723 | 0.748 | 0.361 | 0.315 | 0.386 |
> | 6 | 0.721 | 0.745 | 0.358 | 0.312 | 0.382 |
>
> >**Maximum entropy based view selection**
> | K |  AUC-ROC | AUPR | KS | τ | ρ |
> |---|---|---|---|---|---|
> | 1 | 0.706 | 0.729 | 0.304 | 0.291 | 0.357 |
> | 2 | 0.711 | 0.728 | 0.324 | 0.298 | 0.365 |
> | 3 | 0.705 | 0.720 | 0.312 | 0.290 | 0.355 |
> | 4 | 0.701 | 0.718 | 0.308 | 0.284 | 0.347 |
> | 5 | 0.699 | 0.715 | 0.308 | 0.282 | 0.345 |
> | 6 | 0.699 | 0.712 | 0.308 | 0.281 | 0.344 |
>
> >**Conclusion and Trade-off.**
> >- Selecting the minimum-entropy view (i.e., the most confident view) consistently outperforms selecting the maximum-entropy view when K > 1.
> >- The largest gain occurs when moving from K = 1 to K = 2; beyond K = 2 performance quickly saturates. Thus K=2 provides a strong balance between reliability and cost.
> >- Generating additional views incurs roughly 30 seconds of rendering per extra perspective per motion. Given this cost, we recommend using the engineered human-opt view (i.e., human-opt, K=1), which already achieves performance close to a higher-K regime:
> | human-opt view (K=1) |  AUC-ROC | AUPR | KS | τ | ρ |
> |---|---|---|---|---|---|
> | InternVL3-14B (32 frames) | 0.723 | 0.740 | 0.342 | 0.315 | 0.385 |
>
> > We also release full-body and stick figure motion video demos to an anonymous repo: https://anonymous.4open.science/r/VeMo-322A
>
> ### **Q4, W3. Stability of the VLM-generated scores**
> >Under the VeMo framework (Eq. (1) in the paper), and using InternVL3-14B (32 frames) with the human-opt view, the VeMo scores are effectively deterministic with zero variance. Because Eq. (1) use the soft decoded word distribution generated by VLM to compute Vemo scores.
> >To contrast, deterministic decoding of VLM outputs (by matching discrete output options) yields higher variance.
> | Decoding |  AUC-ROC | AUPR | KS | τ | ρ |
> |---|---|---|---|---|---|
> | VeMo Eq. (1) |  0.723±0.000 | 0.740±0.000 | 0.342±0.000 | 0.315±0.000 | 0.385±0.000 |
> | Deterministic | 0.627±0.006 | 0.743±0.004 | 0.254±0.012 | 0.262±0.013 | 0.262±0.013 |
>
> >**Conclusion.** The VeMo scoring procedure (soft distributional scoring per Eq. (1)) yields stable, zero-variance scores for deterministic input video. The VeMo scores are substantially more reliable than scores obtained via naive deterministic decoding of VLM outputs.

---

> > ### Comment · Reviewer_4br3 · 2025-11-26
> >
> > I appreciate the authors' effort in conducting extensive additional experiments during the rebuttal phase. Since my major concerns regarding efficiency and reliability have been well-addressed, I am upgrading my rating to 6.

---

> > > ### Author Response · Authors · 2025-11-26
> > >
> > > Thank you again for the insightful suggestions and this incredibly kind message!

---

### Official Review · Reviewer_ccbk · 2025-10-31

**Soundness:** 3
**Presentation:** 3
**Contribution:** 3
**Rating:** 6
**Confidence:** 3

**Summary:**

Text-to-motion generation is a fundamental task in our community. This work proposes an evaluation model, VeMo, to provide a reliable assessment for the task. The core idea is to render generated human motions as skinned videos and then use a video-language model (VLM) to evaluate how well the generated motion aligns with the given prompt. Experiments show that VeMo is an efficient and effective metric for measuring prompt–motion alignment.

**Strengths:**

- The paper is well written and easy to follow.

- The proposed evaluation model is solid and well motivated: it leverages a large video–language model to assess how well generated motions align with prompts, rather than relying on reference-motion metrics.

- Comprehensive experiments demonstrate that the model provides an effective and efficient benchmark for evaluating T2M tasks.

- The ablation study offers sufficient analysis to clarify the design choices.

**Weaknesses:**

The paper lacks a comprehensive evaluation of current text-to-motion methods.
Can the authors provide a thorough benchmarking of existing approaches so the community has a clear reference for comparison?
Such a benchmark would allow future work to make direct, convenient comparisons.

**Questions:**

N/A

---

> ### Author Response · Authors · 2025-11-21
>
> Thank you for the thoughtful suggestion and for recognizing the value of our work, we really appreciate the time you took to read and comment.
>
> ### **Benchmark of Existing T2M Approaches.**
> > We agree that a thorough, community-accessible benchmark of current text-to-motion (T2M) methods is important. In response, we have produced a comprehensive evaluation of classic and representative state-of-the-art open-source T2M models and summarized the results in the official comment at the top of our rebuttal. That comment contains the full tables, metrics, and discussion you requested. We hope these results provide a clear reference for future comparisons; the evaluation code will be released so others can reproduce and extend our benchmark.
> >
> >We also release motion video demos from different T2M models to an anonymous repo: https://anonymous.4open.science/r/VeMo-322A
>
> > Thanks again for this constructive recommendation, it helped us strengthen the paper and will, we believe, make comparisons in future work both easier and fairer.

---

> > ### Comment · Reviewer_ccbk · 2025-11-24
> > **Official comment by Reviewer ccbk**
> >
> > Did you include the benchmark table in the revision? Please highlight the newly added results in the revised manuscript (maybe using red color text).

---

> > > ### Author Response · Authors · 2025-11-24
> > >
> > > Thank you again for your helpful reminder! We have included the benchmark table and other results in the latest revision and highlighted all newly added results in red text as requested.

---

> ### Author Response · Authors · 2025-11-24
>
> Thank you for the kind reminder and for your very rapid reply, it’s truly exciting to receive such prompt feedback. We are working around the clock to update the manuscript formatting so we can include the many new results (which will be clearly highlighted) from the experiments we recently completed. We will submit the revised version very soon and truly appreciate your patience and helpful suggestions.

---

> ### Comment · Reviewer_ccbk · 2025-11-26
> **Official comments by Reviewer #ccbk**
>
> I sincerely appreciate the authors' prompt response to my questions. After reviewing the other reviewers' comments and the authors' feedback, I have no further concerns and will maintain my original score.

---

> > ### Author Response · Authors · 2025-11-27
> >
> > Thank you so much for the thoughtful comments and for your endorsement of our work!

---

### Official Review · Reviewer_MQKN · 2025-11-01

**Soundness:** 3
**Presentation:** 3
**Contribution:** 2
**Rating:** 4
**Confidence:** 3

**Summary:**

The paper considers solving the T2M evaluation task by making use of a video language model, and provides a meta-evaluation dataset.

**Strengths:**

1.The paper addresses a critical limitation in T2M evaluation, over-reliance on scarce motion data by proposing VeMo, a novel framework that repurposes VLMs (trained on internet-scale text-vision data) for text-motion alignment assessment.

2.VeMo provides a plug-and-play solution for T2M evaluation—no motion data or human labels are needed for training, making it easily integrable into existing T2M pipelines.

**Weaknesses:**

1.The main experiments only use InternVL3-14B as the core VLM, with limited testing of other open-source VLMs (only 4 models in ablation). This fails to verify VeMo’s robustness across VLM architectures (e.g., video-specialized vs. general multimodal models). Additionally, the benchmark only includes motions from HumanML3D and two T2M models—excluding complex motions or newer T2M models,limiting validation of VeMo’s generalizability .

2.VeMo requires rendering 3D motions into multi-view videos (Blender) and running large VLMs. The paper provides no analysis of computational efficiency or optimization strategies for low-resource scenarios, limiting its practical applicability for real-time T2M evaluation. This will lead to reduced usability.

3.There are too few video demos.

4.The headings in the supplementary materials should be numbered as 1., 2. instead of .1, .2.

**Questions:**

1.The benchmark excludes complex motions and newer T2M models. Could author extend the benchmark to include these, and reevaluate VeMo to confirm its generalizability?

2.Author note VeMo inherits VLM biases but provide no details.

3.VeMo’s computational cost is high. Could you test lightweight VLMs (e.g., InternVL3-8B) or optimize rendering to reduce resources, while quantifying the trade-off between efficiency and evaluation performance?

4.The entropy-based view selection assumes low entropy equals low information loss. Could you provide a deeper explanation for this correlation? Are there edge cases where this strategy fails, and how might you address them?

---

> ### Author Response · Authors · 2025-11-21
>
> Thank you for the careful and constructive feedback, we really appreciate the time you spent reading our work and the thoughtful suggestions that helped us improve clarity. Below we provide concise clarifications and additional results addressing each point raised by the reviewers.
>
> ### **Q1,W1. Clarification on T2M Benchmark and Meta-evaluation Benchmark**
> >We would like to emphasize the different purposes of the two benchmarks. The Text-to-Motion (T2M) benchmark is intended as a broad benchmarking of existing T2M approaches; please see the official comment at the top for our comparisons that include several newer T2M models.
> By contrast, the meta-evaluation benchmark introduced in this work has a focused goal: to construct a fair human-rating reference and to verify the basic evaluation effectiveness of T2M evaluators on a standard set. Our experiments and analyses are framed around this objective, and they demonstrate that the VLM-based T2M score we introduce is substantially closer to human ratings than existing automatic evaluators, which is an important step toward improving T2M evaluation.
>
> >Scaling the meta-evaluation to cover a much wider range of motion complexity and more T2M models is valuable but also labor-intensive and outside the core contributions claimed here. We will explicitly discuss this limitation in the paper and identify it as a clear direction for future work.
>
>
> ### **Q2. Clarification on Details of VLM biases**
> >VeMo inherits the known biases of VLMs, including cultural and other social biases. We already mention this in the Ethics statement; however, we agree that the discussion can be strengthened. We will expand the Ethics section to more clearly describe the potential biases and add citations for readers who want deeper background (e.g., benchmarking work on cultural understanding in VLMs [1]).
> >- [1] Benchmarking Vision Language Models for Cultural Understanding
>
> ### **Q3,W2. Computational Overhead and Efficiency Tradeoff of the VeMo Pipeline**
> >We agree that computational cost is an important practical consideration. Your suggestion to test smaller VLMs (for example, InternVL3-8B) and to optimize rendering is constructive. We have posted an official comment at the top summarizing our initial efficiency experiments and trade-off discussion; please refer to that for preliminary results. We include a clearer analysis of runtime and resource usage, and explicitly note the potential for improvements through both lighter VLMs and parallel rendering pipelines.
>
> ### **Q4. Explanation of Entropy-based View Selection**
> >Our use of entropy for view selection is motivated by prior work linking visual uncertainty to hallucination and to the entropy of model predictions. In particular, studies [2] on hallucination in VLMs show that higher visual uncertainty can amplify language priors and statistical biases, increasing the likelihood of hallucination. Other work [3] has demonstrated a strong empirical relationship between hallucination and semantic-entropy measures. Inspired by these findings, we select low-entropy views: such views tend to yield more reliable and less ambiguous multimodal signals, which improves consistency in evaluation. Our experiments (Section 5.4.2) confirm that choosing low-entropy views produces better average evaluation performance than selecting high-entropy views.
>
> >VLMs may give incorrect high-entropy predictions for input videos, these edge cases are usually considered out-of-distribution for the model [1-2]. This limitation reflects a gap between current VLM video-understanding capabilities and human perception. Addressing these corner cases will require models with more human-like motion perception and higher-fidelity grounding.
>
> >Finally, perfect agreement between any automatic evaluator and human raters is unattainable (see Table 3), since text–motion alignment is often subjective and multimodal. Our primary objective is to narrow the gap between automatic T2M evaluators and human judgment; tackling remaining edge cases is an important direction for future work.
> >- [2] A Survey on Hallucination in Large Vision-Language Models
> >- [3] Detecting hallucinations in large language models using semantic entropy.
>
> ### **W3,W4. Additional Materials & Formatting**
> >We have uploaded additional video demos at anonymous repo: https://anonymous.4open.science/r/VeMo-322A
> >
> >We will also fix the formatting issue in the supplementary materials.

---

### Official Review · Reviewer_7rLa · 2025-11-05

**Soundness:** 3
**Presentation:** 3
**Contribution:** 2
**Rating:** 4
**Confidence:** 3

**Summary:**

This paper proposes a new evaluation metric for text-to-motion generation by utilizing a pretrained video language model. The idea is to leverage the prior knowledge residing in the VLM and render generated motions in different viewpoints for the VLM to determine its consistency with the text. A meta-evaluation benchmark is proposed to evaluate the effectiveness of the proposed metrics.

**Strengths:**

- Utilizing VLM as a prior model to evaluate motion generation is interesting and introduces a new perspective for text-to-motion generation evaluation.
- Experimental results and analysis are performed on the benchmark dataset to validate the proposed evaluation method.
- The writing is good and the paper is easy to follow.

**Weaknesses:**

- The benchmark data is generated by two motion generators, which may contain inductive bias in the motion generator, which may not be the real-world distribution.
- The evaluation metric has its limitations. Since a human annotator only annotates yes or no options. It cannot reveal the extent of failure cases, e.g., the generated motions perform mostly correctly following the text, while with some small mistakes, or the motion is totally wrong.
- The high entropy scores selection strategy in Eqn. 3 only utilizes the highest score single-view for the decision. However, some complex motions may suffer from the overlapped problem in different sub-sequences from a single view, e.g., performing some motion while turning in a circle at the same time. Any strategies to combine information from different views for a better performance?
- The user study only contains two users, which may contain variation.

**Questions:**

Null.

---

> ### Author Response · Authors · 2025-11-22
>
> Thank you for the careful reading and constructive feedback. We truly appreciate the reviewers’ time and insightful comments, they helped us clarify important points and improve the presentation. Below we provide concise responses and supplementary results addressing each concern.
>
> ### **W1. Clarification on Data Generation in Meta-Evaluation**
> >VeMo is a zero-shot evaluator and is not trained on generated motions. As such, it evaluates generated outputs without inheriting the inductive biases of any particular motion generator. This design choice enables fairer, more generator-agnostic comparisons across diverse T2M systems.
>
> ### **W2. Clarification on the Coarseness of the Evaluation Signal**
> >The VeMo score was intentionally designed to answer the most fundamental and objectively measurable question in text-to-motion (T2M): Does the generated motion match the text? Current automatic metrics often struggle to reliably distinguish even coarse-grained “yes/no” alignment, so we focused first on this basic and well-defined problem. We agree that fine-grained degrees of adherence (e.g., mostly correct with small mistakes vs. completely wrong) are important; however, they require richer annotations and more capable evaluation tools. We view fine-grained alignment as an important direction for future work.
>
> ### **W3. Combining Information from Multiple Views**
> >To probe whether multi-view fusion helps, we ran an experiment using videos where each frame contains two synchronized views (one human-optimal view and one randomly rotated view). Evaluation with InternVL3-14B (32-frame) yields:
> | Video used in Vemo | AUC-ROC | AUPR | KS | τ | ρ |
> |---|---|---|---|---|---|
> | Video in two view | 0.716 | 0.740 | 0.348 | 0.306 | 0.374 |
> | Video in human-opt view | 0.723 | 0.740 | 0.342 | 0.315 | 0.385 |
>
> >We find that naively combining multiple views can confuse current VLMs and slightly degrade VeMo performance. This suggests that effective multi-view integration likely requires improvements in VLM video understanding (better temporal and multi-view fusion). We therefore identify multi-view fusion as an important avenue for future work rather than claiming a simple aggregation rule is sufficient today.
>
>
> ### **W4. Clarification on User Study**
> >First, the user study is used only as an independent baseline evaluator rather than as part of the meta-evaluation benchmark construction; it therefore does not bias the benchmark. The user annotations are unseen by other baseline evaluators and are used to analyze agreement between naive users and oracle annotators (see Section 5.4.1).
>
> >Second, we supplement additional user scores here for completeness. These additional results are consistent with Table 3 in the paper: (1) correlation of human evaluation is above 0.6; (2) correlation of the reference-free evaluator is below 0.1; and (3) correlation of VeMo is below 0.4.
> | Method | AUC-ROC | AUPR | KS | τ | ρ |
> |---|---|---|---|---|---|
> | MoBERT (Best alternative) | 0.549 | 0.553 | 0.088 | 0.069 | 0.084 |
> | VeMo (min entropy view) | 0.720 | 0.743 | 0.354 | 0.311 | 0.381 |
> | VeMo (human-opt view) | 0.723 | 0.740 | 0.342 | 0.315 | 0.385 |
> | User-1 | 0.829 | 0.878 | 0.658 | 0.659 | 0.659 |
> | User-2 | 0.835 | 0.876 | 0.670 | 0.677 | 0.677 |
> | User-3 | 0.833 | 0.877 | 0.665 | 0.666 | 0.666 |
>
> >We also report inter-annotator agreement (Krippendorff’s α) between each user and the oracle annotators (values taken from Table 4):
> | IAA (Krippendorff’s α) | Alignment | Naturalness | Faithfulness |
> |---|---|---|---|
> | User-1 | 0.6566 | 0.6376 | 0.7356 |
> | User-2 | 0.6681 | 0.6564 | 0.7896 |
> | User-3 | 0.6563 | 0.6348 | 0.7505 |
>
> >For context, MoBERT [1] (our baseline) also collected human feedback to train a T2M evaluator and reports comparable IAA values for Naturalness/Faithfulness (average IAA ≈ 0.647 and 0.701 respectively). Thus, the agreement levels in our small user study are consistent with prior work and support the reliability of the human baseline we provide.
> >- [1] What is the Best Automated Metric for Text to Motion Generation?

---

### Author Response · Authors · 2025-11-21
**Major Concerns Across All Reviews**

We deeply appreciate the reviewers and the Area Chair for their time and constructive feedback.
Below, we summarize our responses to the main shared concerns, while specific points will be addressed in each reviewer’s thread.

### **Computational Overhead and Efficiency Tradeoff of the VeMo Pipeline.**
>Our full VeMo pipeline consists of three stages: Joint-to-Mesh, Mesh-to-Video, and VLM inference. Converting a rendered 3D mesh into multi-view videos does not increase the Joint-to-Mesh overhead. The measured time and peak RAM for rendering are:
| Video Converter | Per-motion Time / Peak-RAM (Joint-to-Mesh) | Per-motion Time / Peak-RAM (Mesh-to-Video) |
|---|---|---|
| Rendering-to-Body-Model | 182s / 793MiB | 31s / 256MiB |

> The runtime, memory footprint, and evaluation performance of VeMo under different VLM configurations (human-opt view) are:
| VeMo (human-opt view) | Per-Video Time/Peak-RAM | AUC-ROC | AUPR | KS | τ | ρ |
|---|---|---|---|---|---|---|
| InternVL3-14B (32-frame) | 1.597s / 33221MiB | 0.723 | 0.740 | 0.342 | 0.315 | 0.385 |
| InternVL3-14B (8-frame) | 0.415s / 30000MiB | 0.720 | 0.738 | 0.343 | 0.311 | 0.381 |
| InternVL3-8B (32-frame) | 0.889s / 18332MiB | 0.687 | 0.706 | 0.291 | 0.264 | 0.324 |
| InternVL3-8B (8-frame) |  0.233s / 15998MiB | 0.683 | 0.701 | 0.284 | 0.258 | 0.316 |
| InternVL3-1B (32-frame) | 0.295s / 4470MiB | 0.630 | 0.627 | 0.215 | 0.184 | 0.225 |
| InternVL3-1B (8-frame) | 0.084s / 2503MiB | 0.642 | 0.645 | 0.231 | 0.201 | 0.246 |

> For reference, the best-performing reference-free baseline is:
|  | AUC-ROC | AUPR | KS | τ | ρ |
|---|---|---|---|---|---|
| MoBERT | 0.549 | 0.553 | 0.088 | 0.069 | 0.084 |

>**Takeaways.** The primary time bottleneck is rendering, but rendering is highly parallelizable because its peak RAM is low. The primary RAM bottleneck is VLM inference; using smaller VLMs (for example, InternVL3-1B) reduces memory requirements at the cost of modest performance degradation. This trade-off makes VeMo practical for different resource budgets.


### **Benchmark of Existing T2M Approaches.**
>We adopt InternVL3-14B (32-frame) with the human-opt view selection as our VeMo backbone and report VeMo scores on the HumanML3D benchmark covering classic and representative state-of-the-art T2M models to give the community a clear baseline for comparison:
| Method | VeMo↑ | FID↓ | MM Dist↓ | R@1-Precision↑ | R@2-Precision↑ | R@3-Precision↑ |
|---|---|---|---|---|---|---|
|MDM [1]| 0.6171±0.0024  | 0.544±.044 | 5.566±.027 | 0.491±.001 | 0.681±.001 | 0.782±.001 |
|MotionGPT [2]| 0.5723±0.0034 | 0.232±.008 | 3.096±.008 | 0.492±.003 | 0.681±.003 | 0.778±.002 |
|StableMofusion [4]| 0.6528±0.0001 | 0.098±.003 | 2.770±.006 | 0.553±.003 | 0.748±.002 | 0.841±.002 |
|MLD-M [3,5]| 0.6626±0.0002 | 0.073±.003 | 2.810±.008 | 0.548±.003 | 0.738±.003 | 0.829±.002 |
|MotionLCM-V2 [5]| 0.6638±0.0005 | 0.072±.003 | 2.767±.007 | 0.546±.003 | 0.743±.002 | 0.837±.002 |
|Real| 0.6825±0.0000 | 0.002±.000 | 2.974±.008 | 0.511±.003 | 0.703±.003 | 0.797±.002 |
>
>- [1] Human motion diffusion model
>- [2] Motiongpt: Human motion as a foreign language
>- [3] Executing your commands via motion diffusion in latent space
>- [4] StableMoFusion: Towards Robust and Efficient Diffusion-based Motion Generation Framework
>- [5] MotionLCM-V2: Real-time Controllable Motion Generation via Latent Consistency Model

>Notably, FID is a reference-based metric, and its automatic evaluation performance is second only to VeMo (Appendices Table 7); however, FID is a model-level metric calculated on the overall distribution of generated data and cannot be used for point-wise T2M evaluation.

>We observed that several T2M models outperform the ground truth on certain reference-free metrics (e.g., MM Dist and R@K-Precision), which suggests those evaluators can be overfit or “hacked.” To support reproducibility and ongoing benchmarking, we will open-source our out-of-the-box evaluation code and keep the benchmark updated.

---

### Author Response · Authors · 2025-11-25
**PDF Update (Nov 24) — Updated Supplementary Results Following Reviewer's Suggestion**

Dear Reviewers and AC,

Following suggestions, we updated the main paper and Appendices on Nov 24. The newly added results in the revision are highlighted in red.

The update adds more analytical experiments and does not conflict with any original conclusions.

Best,
Authors

---

### Note · Authors · 2026-01-28

I have read and agree with the venue's withdrawal policy on behalf of myself and my co-authors.

---

### Meta-Review · Area_Chair_wfXk · 2026-01-07

**Summary:**

This submission presents a VLM-based evaluator for text-to-motion generation and addresses a meaningful problem. However the core concerns raised by reviewers remain unresolved. The method is only tested on a narrow set of generators, depends on a single VLM model, and relies on a single rendered view despite projection ambiguities. The computational burden also limits practical use and the rebuttal reframes these issues as future work rather than addressing them experimentally. As a result the evaluator is not yet sufficiently validated or demonstrated to be robust or broadly applicable. My recommendation is rejection with encouragement to resubmit once broader validation is available.

**Reviewer Concerns:**

For Reviewer 7rLa, the rebuttal addresses concerns on annotation reliability and multi-view ambiguity by adding an annotator and running two-view experiments, and clarifying that VeMo is generator-agnostic. However, the generalization concern remains unresolved, as the benchmark still relies on a narrow set of generators and retains a coarse binary evaluation signal.

For Reviewer MQKN, expanded baselines, runtime reporting, and clarified offline use partially satisfy questions about coverage and feasibility. however, reliance on a single VLM backbone and high computational cost remain unsolved.

Reviewer ccbk’s sole request for a comprehensive T2M benchmark is fulfilled and acknowledged in the revision, and the reviewer reports no remaining concerns.

Reviewer 4br3 acknowledged the major concerns regarding efficiency and reliability have been well-addressed

**Reviewer Scores:**

After considering the rebuttal, Reviewer 7rLa would likely maintain the original reject-leaning score since the core concerns were not resolved.

Reviewer MQKN would also likely keep the rejection recommendation, as the rebuttal clarified but did not meaningfully alleviate the reviewer’s principal concerns.

Reviewer ccbk’s evaluation would remain unchanged, as the single request was addressed and no additional issues were raised.

Reviewer 4br3 explicitly stated that the rebuttal resolved major concerns and would increase the score into the accept range.

---

### Decision · Program_Chairs · 2026-01-26

Reject